# Anaplastic Large Cell Lymphoma: Molecular Pathogenesis and Treatment

**DOI:** 10.3390/cancers14071650

**Published:** 2022-03-24

**Authors:** Xin-Rui Zhang, Pham-Ngoc Chien, Sun-Young Nam, Chan-Yeong Heo

**Affiliations:** 1Department of Plastic and Reconstructive Surgery, College of Medicine, Seoul National University, Seoul 03080, Korea; zhangxinrui@snu.ac.kr; 2Department of Plastic & Reconstructive Surgery, Seoul National University Bundang Hospital, Seongnam 13620, Korea; r3100@snubh.org

**Keywords:** ALK (anaplastic lymphoma kinase), anaplastic large cell lymphoma (ALCL), breast-implant-associated ALCL, pathogenesis, management

## Abstract

**Simple Summary:**

Anaplastic large cell lymphoma is a rare type of disease that occurs throughout the world and has four subtypes. A summary and comparison of these subtypes can assist with advancing our knowledge of the mechanism and treatment of ALCL, which is helpful in making progress in this field.

**Abstract:**

Anaplastic large cell lymphoma (ALCL) is an uncommon type of non-Hodgkin’s lymphoma (NHL), as well as one of the subtypes of T cell lymphoma, accounting for 1 to 3% of non-Hodgkin’s lymphomas and around 15% of T cell lymphomas. In 2016, the World Health Organization (WHO) classified anaplastic large cell lymphoma into four categories: ALK-positive ALCL (ALK+ALCL), ALK-negative ALCL (ALK−ALCL), primary cutaneous ALCL (pcALCL), and breast-implant-associated ALCL (BIA-ALCL), respectively. Clinical symptoms, gene changes, prognoses, and therapy differ among the four types. Large lymphoid cells with copious cytoplasm and pleomorphic characteristics with horseshoe-shaped or reniform nuclei, for example, are found in both ALK+ and ALK−ALCL. However, their epidemiology and pathogenetic origins are distinct. BIA-ALCL is currently recognized as a new provisional entity, which is a noninvasive disease with favorable results. In this review, we focus on molecular pathogenesis and management of anaplastic large cell lymphoma.

## 1. Introduction

Anaplastic large cell lymphomas (ALCLs) are a group of CD30+ mature T cell lymphomas with the same morphological and immunophenotypic features but showing heterogeneity in their clinical and genetic characteristics [1]. Currently, anaplastic large cell lymphomas of all types represent about 15% of peripheral T cell lymphomas throughout the world [2,3]. The incidence of ALCL is considered to be 0.25 cases per 100,000 people in the United States of America, sharing 3 to 5% of all non-Hodgkin’s lymphomas [4,5,6].

In 1985, anaplastic large cell lymphoma was first recognized by Stein et al. as “anaplastic large cell Ki-1-positive lymphomas”, an aggressive non-Hodgkin’s lymphoma (NHL) suggesting “malignant histiocytosis” in morphology, but sharing features in immunohistochemistry with classic Hodgkin’s lymphoma. This entity was defined as the proliferation of large lymphoid cells that are prone to invading lymph node sinuses and shows high expression of Ki-1 antigen (later named CD30) [7]. In 1988, anaplastic large cell lymphoma was included in the Kiel classification [8]. Next, in 1994, it was incorporated into the Revised European–American Lymphoma (REAL) classification [9], and in 2001, it was included in the WHO classification of hematopoietic and lymphoid tissues [10]. According to the WHO Classification of Tumours of Haematopoietic and Lymphoid Tissues, there are four subtypes of anaplastic large cell lymphoma: ALK-positive ALCL (ALK+ALCL), ALK-negative ALCL (ALK−ALCL), primary cutaneous ALCL (pcALCL), and breast-implant-associated ALCL (BIA-ALCL) [11].

Defined as aggressive systemic lymphomas, ALK+ALCL and ALK−ALCL are distinguishable based on genetic alterations of the ALK gene located on chromosome 2, causing the abnormal expression of ALK protein (Figure 1). ALK, that is, anaplastic lymphoma kinase, is a protein product of the ALK gene located on chromosome 2. The cell signaling pathways and the expression of various genes can be activated by ALK via epigenetic mechanisms, which may stimulate aberrant cell proliferation and growth, and finally promote malignancy [12,13]. Although ALK-negative ALCL is not involved with ALK translocations, other translocations, rearrangements, and mutations related to genes may contribute to its development [14].

Different from ALK+ALCL and ALK−ALCL, primary cutaneous ALCL and-breast implant-associated ALCL are far less aggressive lymphomas presenting only one or very few sites. The prognoses of both are excellent. Generally, pcALCL exhibits skin papules or tumors that typically are limited to the dermis without invading the hypodermis or metastasizing [11]. BIA-ALCL is a complication caused by breast implants. It usually presents in deformation and pains around the breast implants many years after surgical implantation. The complication is localized to the involved breast in most cases.

## 2. ALK-Positive ALCL

### 2.1. Definition

Anaplastic lymphoma kinase-positive anaplastic large cell lymphoma (ALK+ALCL) is a kind of peripheral T cell lymphoma (PTCL) presenting large lymphoid cells with ample cytoplasm as well as pleomorphic nuclei. It is characterized by ALK gene translocation and ALK protein expression, as well as CD30 expression [1].

### 2.2. History

In 1982, the monoclonal antibody Ki-1 (later named TNFRSF8 or CD30) was highlighted when studying the Hodgkin and Reed–Sternberg cells of Hodgkin’s disease [16]. In 1985, Stein et al. described a disease which was called anaplastic large cell Ki-1-positive lymphomas when his team noticed a group of large cell lymphomas suggesting “malignant histiocytosis” in morphology and embodying the high expression of the antigen Ki-1 [7]. Following that, it was widely accepted that CD30-positive anaplastic large cell lymphoma was a separate clinicopathological entity. Then, the revised Kiel classification and the revised European–American lymphoma classification incorporated this disease in 1988 and 1994, respectively [8,9,17]. Later, anaplastic lymphoma kinase-positive anaplastic large cell lymphoma (ALK+ALCL) was recognized as a distinct entity in the 2008 World Health Organization (WHO) classification of malignant lymphomas [18].

### 2.3. Risk Factors

No specific risk factors have been confirmed for anaplastic large cell lymphoma. Currently, the pathogenic factors which can cause non-Hodgkin’s lymphoma in human beings, such as pesticides, fertilizers, the Epstein–Barr virus, and human T cell leukemia/lymphoma virus, have not been proven to relate to the development of anaplastic large cell lymphoma. Furthermore, there is no clear evidence that persistent antigenic stimulation has a role in the development of ALCL [19]. In addition, despite the fact that anaplastic large cell lymphoma is not classified as an HIV-related lymphoma by the WHO, at least 20 instances have been recorded, with rare incidences of ALK expression [20]. Rapid deterioration, diagnostic delay, and poor outcomes are characteristics of anaplastic large cell lymphomas associated with HIV [21].

### 2.4. Clinical Features

In cases of adult and pediatric or adolescent non-Hodgkin’s lymphoma, ALK-positive anaplastic large cell lymphoma accounts for 1–3% and 10–20%, respectively [1,22]. Systemic ALK-positive anaplastic large cell lymphoma primarily develops in young individuals and exhibits a slightly higher incidence rate in males. It manifests as lymphadenopathy that progresses quickly, as well as systemic symptoms such as fevers, excessive sweating, and weight loss. When patients are diagnosed, they are usually in an advanced stage of their illness, showing lymph node enlargement and systemic symptoms. Extranodal involvement is common and present in the majority of cases, which includes skin (26%), bone (14%), soft tissues (15%), lung (12%), and liver (8%) [23,24,25,26].

### 2.5. Molecular Pathogenesis

The ALK translocation, which was first discovered in 1994, is the characteristic genetic alteration in ALK-positive ALCL [27]. Most of the cases (approximately 80%) show a chromosomal translocation t(2;5), which leads to the fusion of the ALK gene at 2p23 to the nucleophosmine (NPM) gene at 5q35. The existence of a chimeric ALK fusion protein plays a crucial role in the progression of these lymphomas [27]. Apart from that, other translocations associated with ALK also occur as listed below. In total, 13% of cases involve tropomyosin 3 (TPM3: 1q25), 1% of cases involve IMP cyclohydrolase/5 aminoimidazole-4-carboxamide ribonucleotide formyltransferase (ATIC: 2q35), and 1% of cases involve TRK-fused gene (TFG: 3q12.2). As for the other translocations, the rate of occurrence is lower than 1%, including myosin heavy chain 9 (MYH9: 22q11.2), clathrin heavy chain (CLTC: 17q23), tropomyosin 4 (TPM4: 19p13.1), moesin (MSN: Xq11-12), ring finger protein 213 (RNF213: 17q25) [28,29,30,31,32,33,34,35,36,37,38,39,40]. All of the mutant translocations result in constitutive ALK-tyrosine kinase activation, which is an important change in this lymphoma [41,42,43].

The ALK component of the NPM-ALK protein, which contains the ALK receptor’s catalytic domain, is autophosphorylated by reciprocal ALK tyrosine kinase activity, resulting in intense and long-lasting activation [44,45]. Constitutively active ALK fusion proteins induce tumorigenesis through phosphorylation as well as activation of signaling pathways including STAT3, STAT5, PLC γ, mTOR, EKR, PI3K, MEK, and AKT1 [46,47,48,49,50] (Figure 2). The carcinogenic activity of STAT3 is mediated via its control of a variety of target genes involved in the cell cycle, death, immune system, vasculature, and metabolism [51]. STAT3 plays a key role in the formation of ALCL by mimicking physiological pro-growth signals, such as the IL2 and TCR signaling pathways, promoting the proliferation and survival of tumor cells [52,53]. Recent research has shown that interferon regulatory factor 4 (IRF4) is the STAT3 key target gene that promotes neoplastic cell survival by activating the transcription factor MYC. Nevertheless, independently of IRF4 genetic mutations (translocations or overexpression), it is proven that IRF4 can induce the survival of ALCL neoplastic cells [54,55]. This process is described as an autoregulatory circuit in myeloma cells, with IRF4 directly targeting MYC while also serving as a direct target of MYC transactivation [56].

Moreover, other mechanisms involving MYC may also play a significant role in ALCL. For example, the increased expression of Basic Leucine Zipper ATF-Like Transcription Factor 3 (BATF3) mediated by the JAK-STAT signaling pathway can result in the activation of MYC, which can affect the proliferation and apoptosis of cells and contribute to the formation of cancer [58]. Other mechanisms involved in inducing lymphomas have been reported, including making genome unstable via phosphorylation of DNA mismatch repair proteins [59]; boosting angiogenesis by upregulating hypoxia-inducible factor 1-alpha and vascular endothelial growth factor A [60,61]; advancing metastasis via reversion to stem cell-like phenotype with the overexpression of SOX2, SALL4, and TWIST1 [62,63,64]; promoting tumor inflammation by the increment of cytokines which are involved in adaptive immunity, such as interleukin 21 [65]; helping escape from the immune response through immunosuppression that is mediated by the overexpression of interleukin 10, transforming growth factor-beta, and programmed death-ligand 1 [66,67]; and resisting apoptosis through the upregulation of antiapoptotic protein myeloid cell leukemia 1 and BCL2-related protein A1 [68,69,70].

Moreover, it is worth noting that there exist some cases showing the expression of CD56. According to retrospective analysis, the proportion of systemic ALK-positive ALCL presenting CD56+ is about 20%. The cases which underwent TCR gene rearrangement detection exhibited monoclonal TCR gene rearrangement, with the same phenotype of TCR-γ+/β+/δ− [71].

### 2.6. Management

For the management of newly diagnosed patients, multiagent anthracycline-containing regimens remain the gold standard. Currently, CHOP is the most extensively utilized regimen. Multiple studies have found out that overall and progression-free survival (PFS) rates are over 70% and 60%, respectively, after five years. In a major meta-analysis, the German High-Grade non-Hodgkin’s Lymphoma Study Group (Deutsche Studiengruppe Hochmaligne Non-Hodgkin Lymphome) investigated the addition of etoposide to CHOP, often in the form of CHOEP (etoposide, 100 mg/sqm on days 1, 2, and 3). A separate evaluation presented the results of 320 PTCL patients [72]. In younger ALK+ALCL patients (*n* = 78) with normal LDH at the time of diagnosis, adding etoposide to CHOP(-like) regimens enhanced overall response rates and gave superior event-free survival (EFS) (3-year EFS of 91% vs. 57% in CHOEP vs. CHOP-treated patients, respectively). The Swedish Lymphoma Registry confirmed that adding etoposide to CHOP improved PFS in patients with PTCL who were younger than 60 years old [73]. In the study, ALK+ALCL was found in 68 of 755 PTCL patients; no histology-specific analysis was performed. Continuous infusion of chemotherapeutic drugs may enhance results in patients with highly proliferative cancers. In ALCL patients with high-risk traits, dose-adjusted EPOCH (etoposide, cyclophosphamide, doxorubicin, vincristine, and prednisone) has been tested. After a median follow-up of more than 12 years, median survival for both ALK+ and ALK− patients was not reached, with a 10-year OS rate of 75% [74]. Furthermore, a pooled analysis of 263 patients demonstrated that etoposide-based induction was associated with improved 5-year PFS (83% versus 62%) and 5-year OS (93% versus 74%) using etoposide versus non-etoposide regimens. In patients ≤ 60 years (*n* = 232), the respective 5-year PFS and OS were 81% versus 65% and 92% versus 77%. However, among the 31 patients > 60 years, given the additive toxicity, only 6 received etoposide with their induction and 25 did not. The respective 5-year PFS and OS were 100% versus 42%, and 100% versus 53% [75]. According to these findings, CHOEP should be chosen for initial therapy in younger patients with ALK+ALCL, while CHOP and DA-EPOCH should be retained for older or less fit patients.

While the majority of ALK+ALCL patients are expected to be treated with the frontline treatment described above, up to 30–40% patients experienced relapses. For these kinds of patients, universally used combinations are ICE (ifosfamide, etoposide, carboplatin) [76], DHAP (dexamethasone, cisplatin, high-dose cytarabine), ESHAP (etoposide, solumedrol, high-dose cytarabine, carboplatin) [77], GemOx (gemcitabine, oxaliplatin) [78], and GDP (gemcitabine, dexamethasone, cisplatin) [79], which are projected to achieve complete remission in almost half of the patients. Crizotinib, an ALK inhibitor, has been shown to have very excellent effects in clinical trials, notably in the juvenile population [80,81]. In a trial conducted by the Children’s Oncology Group (COG), 21 out of 26 pediatric patients showed a complete response to crizotinib as a first-line treatment for ALK inhibition [82]. Unfortunately, patients with ALK-positive lymphoma who stopped taking crizotinib experienced a sudden relapse [83]. Several clinical trials with crizotinib, lorlatinib, and ceritinib present promising preliminary results. Despite the preliminary efficacy of ALK kinase inhibition in ALK+ALCL, resistance mutations have been discovered [84], reducing ALCL cell sensitivity to several ALK inhibitors [85]. Nevertheless, the recent approval of brentuximab vedotin, a highly active immunoconjugate, may have put standard management paradigms into question. The treatment of high-dose consolidation with autologous hematopoietic cell transplantation is recommended for patients with the chemotherapy-sensitive disease and is expected to result in a cure or long-term remission in half of the patients [86,87,88]. Those patients with chemotherapy-refractory disease should participate in clinical trials or be treated with novel agents if available. If possible, the transplantation of allogeneic hematopoietic cells should be attempted for those patients who have achieved exceptional partial or complete remission [89,90].

## 3. ALK-Negative ALCL

### 3.1. Definition

Anaplastic lymphoma kinase-negative anaplastic large cell lymphoma (ALK−ALCL) is a CD30-positive T cell non-Hodgkin’s lymphoma. It presents a proliferation of pleomorphic large lymphoid cells with a morphology similar to ALK-positive ALCL, but no chromosomal translocations involving the ALK gene (Figure 3).

### 3.2. History

The history of ALK-negative anaplastic large cell lymphoma was the same as ALK-positive anaplastic large cell lymphoma until 1994, when Stephan Morris, Thomas Look, and colleagues at Saint Jude Children’s Hospital in Memphis identified the genes involving the chromosomes, such as the chromosomal precise location for chromosomes at t2 and t5, the short arm of the chromosome at p23, and the long arm of the chromosome at q35. It was called anaplastic lymphoma kinase (ALK) [27]. Previously, ALK−ALCL had a short-term existence. However, recently it was categorized by the World Health Organization [92,93].

### 3.3. Risk Factors

There is no particular risk factor that has been clearly identified for ALK−ALCL. Currently, there are not enough data to establish an association between ALCL and inherited immunological deficiency disease or immunocompromised status [94]. Furthermore, unlike other T cell lymphomas, ALK-negative anaplastic large cell lymphoma does not show a close relationship with infection caused by the Epstein–Barr virus (EBV). Only exceptional EBV+ cases have been reported in the literature [95,96].

### 3.4. Clinical Features

Most patients with ALK-negative anaplastic large cell lymphoma are adults aged between 40 and 65, with males being slightly more prone to developing the disease than females (M:F ratio 1.5:1) [95]. In ALK-negative ALCL, about 50% of cases show the involvement of the lymph nodes, while extranodal involvement is less frequent [97,98]. Extranodal sites are relatively common in the skin, soft tissue, liver, and lungs. As for rarely involved sites, they include the oropharynx, gastrointestinal tract, orbit, brain, and testes [99,100,101,102,103,104,105,106,107]. Skin lesions usually present as papules, nodules, or tumors. The involvement of other organs is usually shown as a mass. The presentation of leukemia is very rare in ALK−ALCL, foreshadowing a poor outcome [108,109,110,111].

### 3.5. Molecular Pathogenesis

Similar to ALK-positive anaplastic large cell lymphoma, the clonal rearrangements of the T cell receptor genes can be detected in most ALK-negative anaplastic large cell lymphoma cases. Furthermore, comparative genomic hybridization revealed that ALK-negative anaplastic large cell lymphoma differs from ALK-positive anaplastic large cell lymphoma and other peripheral T cell lymphomas in terms of genetic characteristics (Figure 4) [112].

JAK1 and STAT3 mutations are thought to be linked to lymphoma pathogenesis, as demonstrated in Figure 5A. These mutations cause constitutive activation of STAT, which then translocates to the nucleus and activates many genes, resulting in the expression/activity of the molecules indicated, as well as a T cell lymphoma (TCL) phenotype. Through pSTAT3, chimeric fusion genes with simultaneous transcriptional and kinase activity (ROS1, TYK2) can also maintain the ALCL phenotype. The JAK/STAT pathway is only activated in circumstances when there is no DUSP22 rearrangement.

In addition, clonal chromosomal translocations of the DUSP22-IRF4 locus on 6p25.3 (DUSP22 rearrangements) have been observed in around 30% of systemic ALK-negative anaplastic large cell lymphomas. The most common rearrangement is t(6;7)(p25.3;q32.3), which transposes the DUSP22 locus with non-genic material on chromosome 7. Nevertheless, DUSP22 rearrangements present not only in ALK-negative anaplastic large cell lymphoma, but also in primary cutaneous anaplastic large cell lymphoma and lymphomatoid papulosis [114,115,116,117,118,119,120,121,122]. In vitro studies have demonstrated that DUSP22 can inhibit the signaling induced by T cell receptor genes and c-Jun N-terminal kinase; therefore, it is also named “JNK pathway-associated phosphatase” (JKAP) [123,124] (Figure 5B). DUSP22 can also suppress the STAT3 activation mediated by interleukin 6. In addition, DUSP22 enhances the phosphorylation and activation of STAT3 in cell lines when silenced by siRNA, which confirms the function of DUSP22 in this disease to a certain extent [125]. Declined expression of DUSP22 in t cell lymphomas has been discovered, which suggests that this phosphatase plays a role similar to a tumor suppressor gene in ALK-negative anaplastic large cell lymphomas [114].

Another common alteration is the rearrangement of the TP53 homolog, TP63 (3q23), most frequently in the form of inv(3)(q26q28), which gives rise to the fusion of TBL1XR1/TP63. These cases usually provoke poor prognoses and further study is necessary for these kinds of cases because of the rarity.

### 3.6. Management

For initial-stage ALK−ALCL therapy, CHOP is still the most widely used regimen, but the results are not satisfactory. A study showed that the rate of general palliation is 70–80%, while only 50% of the absolute palliation rate exists, as well as 30–50% of patients living within 5 years of treatment [6,99]. Mosse et al. found that the treatment of 113 ALK−ALCL patients with a combination of etoposides resulted in 61% of 3-year EFS, while the rate of EFS was only 48% with patients without treatment of etoposide [81]. In addition, the study conducted by Ellin et al. showed superior outcomes with the addition of etoposide to CHOP in PTCLs without histology-specific analysis [73]. In a prospective phase II experiment published by the NCI scientists, dose-adjusted EPOCH showed promising results in both ALK+ and ALK−ALCL. The long-term progression-free survival rate, at 72%, was the greatest in any of the prospective ALCL studies [74].

In the frontline setting, more intense regimens were also investigated. The MD Anderson Cancer Center (MDACC) has released the results of a phase II trial that tested an aggressive hyper-CVAD/MA treatment (hyper-fractionated cyclophosphamide, pegylated doxorubicin, vincristine, dexamethasone alternating with high-dose methotrexate and cytarabine) [126]. Although the CR rate reached 83 percent, patients with ALK ALCL had a median progression-free survival of just 7.5 months and a 3-year PFS of 43 percent, which was not significantly different from previous results with CHOP(-like) regimens. Regardless of age, the majority of ALK−ALCL patients were treated using dose-adjusted EPOCH or CHOEP procedures as initial therapy.

Targeted therapy for ALK−ALCL is in research currently. Ruxolitinib, a JAK1/2 inhibitor, was confirmed to be effective in vivo. The data suggested that cytokine receptor signaling is required for the survival of tumor cells, even in the presence of JAK1/STAT3 mutations. Thus, JAK inhibitor therapy is helpful for ALK−ALCL management [127].

## 4. Primary Cutaneous ALCL

### 4.1. Definition

Primary cutaneous anaplastic large cell lymphoma is a rare type of anaplastic large cell lymphoma. According to the 2016 definitions of the WHO and the European Organization for Research and Treatment of Cancer (EORTC), it is incorporated in the primary cutaneous CD30 lymphoproliferative disorders [128,129]. Patients with primary cutaneous anaplastic large cell lymphoma usually exhibit solitary or few clustered tumors. In histology, it is defined by a large number of CD30-positive anaplastic cells with a T cell or null-cell phenotype in the field [130,131,132].

### 4.2. History

In 1989, primary cutaneous anaplastic large cell lymphoma was first described by Berti et al. in a case which was misdiagnosed as cutaneous metastasis [133].

### 4.3. Risk Factors

It is thought that various immunomodulatory drugs can induce primary cutaneous anaplastic large cell lymphoma in patients, including the fingolimod used to treat relapsing-remitting multiple sclerosis, as well as the tumor necrosis factor (TNF) blocker adalimumab used to treat various autoimmune diseases [134,135].

### 4.4. Clinical Features

Primary cutaneous anaplastic large cell lymphoma is encompassed in CD30-positive lymphoproliferative disorder, showing a relatively good prognosis before progressing to an advanced stage [131]. It accounts for around 9% of the diagnoses of all cutaneous T cell lymphomas (CTCLs). Most patients with primary cutaneous anaplastic large cell lymphoma are aged between 50 and 70; however, there have been reports of juvenile and congenital instances as well [136,137].

Clinically, primary cutaneous anaplastic large cell lymphoma most commonly affects the trunk as well as the face. Most of the patients exhibit rapidly growing large (>2 cm) nodules or papules, lasting for at least 3 to 4 weeks [138,139,140]. They can be single or sometimes grouped, with ulceration and a reddish-brown color (Figure 6). Sometimes, primary cutaneous anaplastic large cell lymphomas show spontaneous regression; however, about 50% of the cases will recur [130,132,137,139,141,142,143,144]. It is rare that primary cutaneous anaplastic large cell lymphoma progresses to a systemic disease [4]. Several pieces of research have demonstrated that the involvement of leg/limb induced by primary cutaneous anaplastic large cell lymphoma can lead to poor outcomes, which is more likely to cause extensive and multiple skin lesions and regional nodal involvement [145,146,147,148].

### 4.5. Molecular Pathogenesis

According to the research, most cases of primary cutaneous anaplastic large cell lymphoma show clonal TCR gene rearrangements while lacking the expression of TCR proteins [129,149]. The results of comparative genomic hybridization studies demonstrate that almost two-fifths of patients have chromosomal anomalies in the chromosomal areas coding for NRAS (1p13.2), MYCN (2p24.1), RAF1 (3p25), FGFR1 (8p11), CTSB (8p22), FES (15q26.1), and CBFA2 (21q22.3) [150]. Moreover, gains in 7q31 and 17q and losses in the 6q16–6q21, 6q27, and 13q34 regions have also been detected [151].

In addition, it is believed that DUSP22 is a kind of tumor suppressor, which is in agreement with the lower DUSP22 expression observed in the cases of primary cutaneous anaplastic large cell lymphoma with DUSP22-IRF4 translocations [152]. In 2013, FISH analysis was performed by Karai et al. to describe a rearrangement of IRF4 and DUSP22 on the 6p25.3 locus. Furthermore, 6p25.3 rearrangements can also be used to distinguish other anaplastic large cell lymphomas from primary cutaneous anaplastic lymphoma [119].

### 4.6. Management

Either excision or local radiation can be used in the treatment of primary cutaneous anaplastic large cell lymphoma, and local radiation is considered more appropriate for primary cutaneous anaplastic large cell lymphoma associated with increased tumor load [132,153]. As for systemic chemotherapy, it is generally applied for cases with a greater extent of involvement [153]. In several cases of refractory primary cutaneous anaplastic large cell lymphoma, allogeneic and autologous stem cell transplantation have been tried, while the outcomes are variable [146,154]. The risks associated with stem cell transplantation mean that it is normally reserved for patients who are in good health but have not been cured by other treatments [155].

Targeted therapies are also being studied. Duvic et al. reported that partial or complete responses have been observed in patients with PC-ALCL and LyP after the application of SGN-30, an anti-CD30 monoclonal antibody [156]. A complete response to brentuximab vedotin, a monoclonal antibody to CD30 linked to the cytotoxic compound monomethyl auristatin E, has also shown efficacy in PC-ALCL. As presented in Figure 7, this drug enters the cell by binding to CD30, and then the resulting vesicle fuses with lysosomes, culminating in proteolytic cleavage of the dipeptide linker and the appearance of free MMAE molecules, which block tubulin polymerization of the cellular cytoskeleton and stop pcALCL tumor cell growth. Gamma-secretase (γ-secretase) inhibitors prevent the liberation of intracellular NOTCH1 (ICN1) from membrane-tethered heterodimeric NOTCH1 protein. This leads to the tumor cell nuclear factor-kB (NFKB) pathway being downregulated, and survival genes being inactivated. According to in vitro study, JAK1/2/3 inhibitors are efficient at controlling the development of pcALCL cells. Oncogenic JAK1 or STAT3 mutations are linked to hyperactive pSTAT3 in pcALCL with an NPM1-TYK2 gene fusion and oncogenic STAT3 activation in this pathway. Furthermore, anti-ALK drugs such as crizotinib, alectinib, and ceritinib may downregulate the STAT3 pathway in pcALCL patients with ALK rearrangements, resulting in tumor cell death. IPH4102 is a humanized monoclonal antibody that targets the KIB3DL2 cellular receptor (CD158K). In advanced pcALCL, this receptor has been found to be overexpressed. IPH4140 binds to the CD16 activating receptor in tumor cells and induces cell lysis through antibody-dependent, cell-derived cytotoxicity mediated by NK cells. At the epigenetic level, inhibitors of histone deacetylase (HDAC) and demethylating medicines have demonstrated some effectiveness in inducing cell-cycle arrest, differentiation, and/or death in tumor cells. Nonetheless, peripheral neuropathy is a side effect related to brentuximab vedotin in up to 65% of patients [156,157,158,159].

However, the application of this kind of targeted therapy is restricted by its adverse side effects. No matter what the prognosis is, closely monitoring the patients is necessary due to the potential risk of dissemination.

## 5. Breast-Implant-Associated ALCL

### 5.1. Definition

Breast-implant-associated anaplastic large cell lymphoma is a complication of breast implants that develops many years after surgical implantation. It is a type of T cell lymphoma that presents similar morphological and immunophenotypic features to systemic ALK-negative ALCL. However, the survival rate of BIA-ALCL is higher than systemic ALCL. There are two distinct subtypes of BIA-ALCL: “effusion-only”and “mass-forming” disease. BIA-ALCL may spread to the adjacent breast parenchyma or regional lymph nodes; more rarely, it may spread systemically.

### 5.2. History

In 1997, BIA-ALCL was first described by Keech et al. when they observed a case of ALCL associated with a saline-filled breast implant [160]. Since then, over 900 cases have been reported globally. In January 2011, the United States Food and Drug Administration (FDA) announced a potential association between breast implants and the development of ALCL. Presently, BIA-ALCL is recognized as a cancer of hematological and lymphoid tissues as well as breast tumors by the WHO [1,161].

### 5.3. Risk Factors

In the United States of America, a retrospective study reported that from 1996 to 2015, the risk of developing BIA-ALCL in individuals with breast implants was 67.6 times that of primary breast ALCL in ordinary people. In addition, textured-surface breast implants were more likely to cause BIA-ALCL than smooth-surface breast implants [162]. In July 2019, the FDA requested a recall of textured-surface breast implants owing to the data indicating that there is a relation between BIA-ALCL and textured-surface breast implants [163].

### 5.4. Clinical Features

Generally, the clinical features of BIA-ALCL are indolent. In most cases (>80%), BIA-ALCL is confined to the capsule around the breast implant [164]. If this lymphoma is restricted to the capsule, the problem will usually be solved after removing the implant with its capsule [165]. The involvement of the lymph nodes has a dramatic impact on the prognosis of BIA-ALCL because it means that there is disease progression or that the disease is clinically aggressive. The survival rate of cases without nodal involvement is 97.9%, while that of lymph node-involved cases decreased significantly to 75% [166].

### 5.5. Molecular Pathogenesis

The molecular pathogenesis of BIA-ALCL is still poorly established. Most of the evidence indicates that chronic inflammation and long-term immune stimulation caused by breast implants may cause the disease. Various theories have been proposed, including immune response to breast implants, subclinical bacterial infection, and genetic predisposition.

#### 5.5.1. Chronic Inflammation and Immune Response

Inflammation is an important stage of capsule formation. During this process, different cytokines and cells are recruited and activated. The infiltration of T lymphocytes and monocytes mainly occurs close to the surface of biomaterials and results in the secretion of IL-4, IL-13, and other factors [167,168]. These secreted factors can affect the activity of macrophages and promote the formation of foreign body giant cells (FBGCs). However, the recruitment of T lymphocytes around the breast implants may cause sensitization of T lymphocytes to biomaterials or self-proteins, eventually leading to antigen response and processing. The continuous response of antigens and chronic inflammation can induce immune dysregulation and aberrant cell proliferation. Furthermore, the immune response involves activated T-helper cells (Th1/Th17) (Figure 8). Th17 detects large extracellular pathogens and recruits neutrophils, while Th1 cells recruit macrophages. Macrophages take up bacteria, of which cell-wall components can induce the secretion of interleukin (IL)-1 that activates neutrophils and the production of IL-8. Macrophages also produce IL-12 and IL-6, which drive the maturation of TH1 and Th17 cells, respectively. Furthermore, IL-6 can activate the Janus kinases (JAK)/signal transducer and activator of transcription (STAT) signaling pathway and ultimately give rise to cell proliferation and ALCL.

In addition, the fibrous capsule surrounding breast implants may facilitate tumor escape from the immune system. In addition, the fibrous capsule surrounding breast implants may interfere with the recognition of tumors of the host’s immune system and ultimately lead to antigenic escape. The significance of the immune system in the inhibition of tumors is well confirmed in the research showing that immune-deficient hosts are more likely to develop malignancies compared to normal hosts. It is worth noting that the majority of these malignancies are lymphomas [170].

The existence of fibrous capsules can offer friendly microenvironments to tumors, which promote the development of tumors to a certain extent by the following methods: (1) inhibiting immune-infiltration through down-regulating adhesion molecules and providing physical barriers to antigens [171]; (2) strengthening the expression of immunosuppressive factors from tumors [172].

#### 5.5.2. Subclinical Bacterial Infection

As ducts and glands of the breast communicate with the external environment, it is regarded as a ‘clean-contaminated’ surgical site. The results of scanning electron microscopy have shown that the ratio of Baker grade III/IV capsular contractures which are associated with a biofilm of coagulase-negative bacteria is three times higher than that of mild or no contractures [173]. Moreover, a large number of gram-negative bacteria and various microbiomes can be observed in samples from patients with BIA-ALCL.

It is thought that some cancer cases can be attributable to infections. However, any direct association between bacterial infection and ALCL has not yet been proven. Therefore, it is a kind of theoretical possibility that BIA-ALCL is induced by pathogens.

#### 5.5.3. Genetic Alterations

Cytogenetic studies of BIA-ALCL cases have not shown the presence of chromosomal aberrations related to other lymphomas, including systemic ALK-negative ALCL and primary cutaneous ALCL [174]. However, a few cases have shown genetic alterations, including the gain of 19p and the losses of chromosomes 1p, 10p, and 20p [164,175]. The gain of 19p, the region where a Janus kinase is encoded, may help to clarify the pathogenesis of this lymphoma, due to the involvement of this protein in the phosphorylation of STAT1 and STAT3 [164,176].

Unlike systemic ALK-negative ALCL and primary cutaneous ALCL, the rearrangements of ALCL-associated genes such as ALK, DUSP22, and TP63 have not been detected in BIA-ALCL [177]. The potential molecular driving factors of BIA-ALCL comprise the activation of the JAK/STAT signaling pathway and the dysregulation of TP53 and MYC. The gene mutations associated with JAK/STAT are the most common in this lymphoma [178]. The genetic alterations involved with the JAK/STAT signaling pathway have also been described in systemic ALCL (ALK-positive and ALK-negative) and primary cutaneous ALCL [177]. The activation of the JAK/STAT signaling pathway through STAT3 phosphorylation is quite common in BIA-ALCL. STAT3 mutations can be found in 26% of cases [179,180]. The predominant point mutation (67%) is the STAT3 S614R, which can affect the SH2 domain, resulting in the activation constitutive STAT3 and the phosphorylation of proteins [177]. Around 13% of cases exhibit JAK1 mutations. To date, the JAK1 G1097V mutation is the only one that has been reported in BIA-ALCL. The mutation can cause an increase of protein function and excessive STAT3 phosphorylation. Due to the formation of a positive feedback loop in JAK-STAT mutations, the JAK3 germline mutation V722I is considered a genetic predisposing factor for BIA-ALCL [176,181]. All these mutations can facilitate cell proliferation and oncogene activation, ultimately leading to clonal expansion and tumor development. Additionally, they can also promote the proliferation and growth of lymphoma cells in the microenvironment around the breast implant, which has been demonstrated in cancers associated with chronic inflammation (Figure 9) [182].

### 5.6. Management

Most of the information on the management of BIA-ALCL is from case reports. According to the reported cases, implant removal and capsulotomy, lymph node dissection, chemotherapy, and radiotherapy are the methods that have been applied in clinical treatment.

It is recommended that all patients with BIA-ALCL undergo surgical resection, including breast implant explantation, total capsulectomy, and removing the associated mass with negative margins. The overall survival rate of surgical resection is higher than other interventions, such as chemotherapy or radiotherapy [184]. In addition, it is recommended that the contralateral implant should be removed because of the risk of bilateral disease. Otherwise, the breast implant capsule can drain in multiple regional lymph nodes; thus, a sentinel lymph node biopsy cannot confirm the existence of metastasis. The dissection of axillary lymph node dissection is supposed to be considered when clinical examination or imaging suggests lymph node involvement. All efforts should be made to achieve complete surgical resection so as to prevent local recurrence.

For localized BIA-ALCL, it is recommended for patients to undergo a complete surgical dissection. For locally aggressive disease, complete surgical excision followed by adjunctive therapy is advisable. Patients with stage II or higher cancer, incomplete surgical resection, and biopsy-proven recurrent disease should seek adjuvant therapy. For residual, positive margins, or unresectable localized disease, adjuvant involved-field radiation with a dosage of 24 to 36 Gy is recommended [185]. Systemic chemotherapy is necessary if the disease has progressed to an advanced stage. The most widely used regimen is an anthracycline-based chemotherapy regimen (CHOP (cyclophosphamide, doxorubicin, vincristine, and prednisone)) with or without etoposide, which is similar to that applied for systemic ALCL. Furthermore, brentuximab vedotin is an anti-CD30 antibody–drug combination that has been licensed for the treatment of CD30-positive lymphoid malignancies, such as Hodgkin’s lymphoma, anaplastic large cell lymphoma, and cutaneous CD30-positive lymphoproliferative diseases. Brentuximab was supplemented with CHOP in the ECHELON-2 clinical study. When compared with CHOP treatment in individuals with peripheral T cell lymphomas, this regimen showed a longer-lasting remission and improved overall survival rate [186,187,188]. Dupilumab, a fully human monoclonal antibody that targets IL-4/IL-13 signaling, has shown promise in clinical studies for severe asthma [189] and atopic dermatitis [190], and it may be worth considering for BIA-ALCL treatment. Although IL-6 has been proposed as a driving cytokine in BIA-ALCL, inhibition of the cytokine had little effect on in vitro growth. Clinical trials using tocilizumab, a humanized monoclonal antibody that targets IL-6R, have shown promise in the treatment of chronic inflammatory, autoimmune, and lymphoproliferative illnesses. While IL-6-targeted therapy has not been shown to be effective in the treatment of overt cancers, it may be useful in the development of BIA-ALCL. Another strategy targets the downstream effects of IL-6 signaling, specifically, JAK/STAT signaling. Ruxolitinib, a JAK1/2 inhibitor, is effective in suppressing the proliferation of tumor cells and benefits the patients with this lymphoma [65].

## 6. Conclusions

Anaplastic large cell lymphoma is an uncommon form of T cell lymphomas. It is crucial to be familiar with all the subtypes of ALCL in terms of distinct mechanisms, treatment, and prognosis. In addition, aberrant histologic variants or cases with a “null” phenotype can be overlooked or confused with a variety of other hematopoietic and non-hematopoietic malignancies. Therefore, it is necessary to be aware of the clinical features of ALCL. The recognition of alterations involving genes and JAK/STAT signaling pathway can assist us to stratify these cases into those with a better or worse prognosis and optimize the treatment. Currently, the anthracycline-based chemotherapy regimen is the most widely used therapy throughout the world. The addition of Etoposide to the CHOP regimen has been confirmed as able to improve the PFS and OS significantly. Furthermore, the application of targeted therapies based on molecular mechanism have been studied, and brentuximab vedotin as well as other inhibitors are helpful for treatment. The understanding of the molecular pathogenesis and management of anaplastic large cell lymphoma will have a profound and positive impact on the study of this disease, which may help to make a progress in the field of oncology.

## Figures and Tables

**Figure 1 cancers-14-01650-f001:**
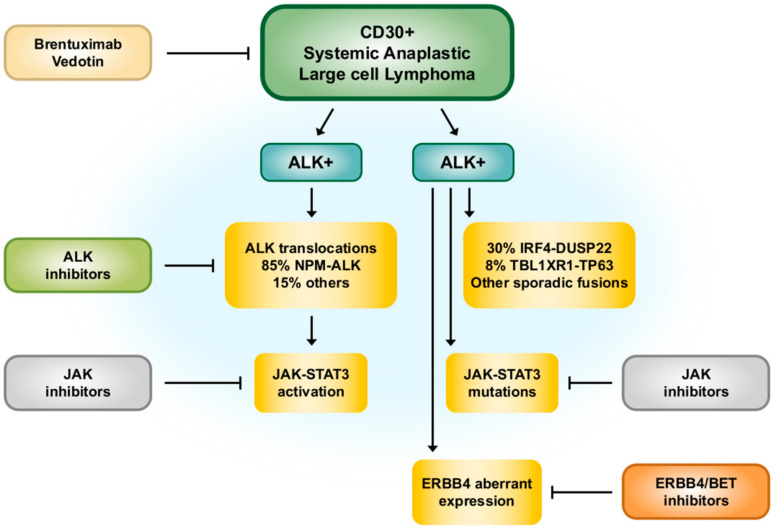
Schematic presentation of different entities of systemic Anaplastic Large Cell Lymphomas (ALCL). ALK+ALCL is well defined by localization of ALK. ALK−ALCL was presented with different cytogenetic subsets. The pharmacological inhibitors in different stages are depicted with their respective target. Reproduced with permission from [15].

**Figure 2 cancers-14-01650-f002:**
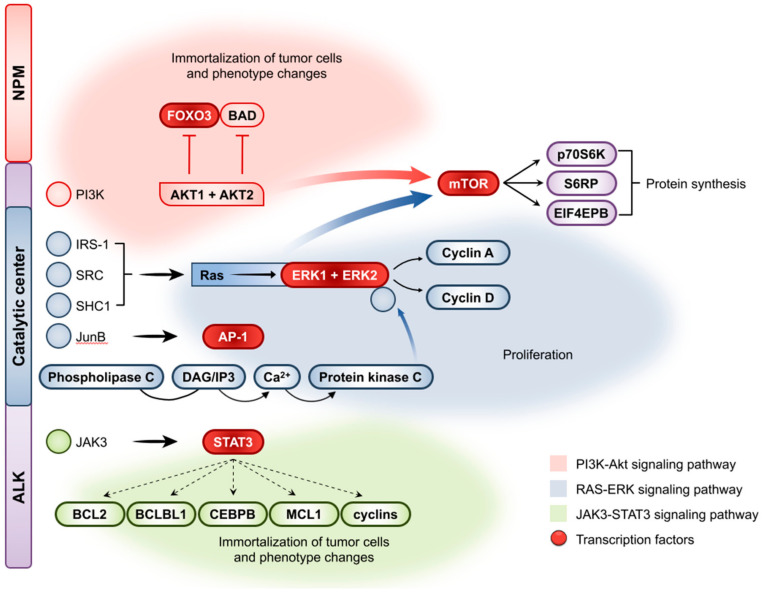
Schematic representation of anaplastic lymphoma kinase recombinant proteins on intracellular signaling pathways. Reproduced with permission from [57].

**Figure 3 cancers-14-01650-f003:**
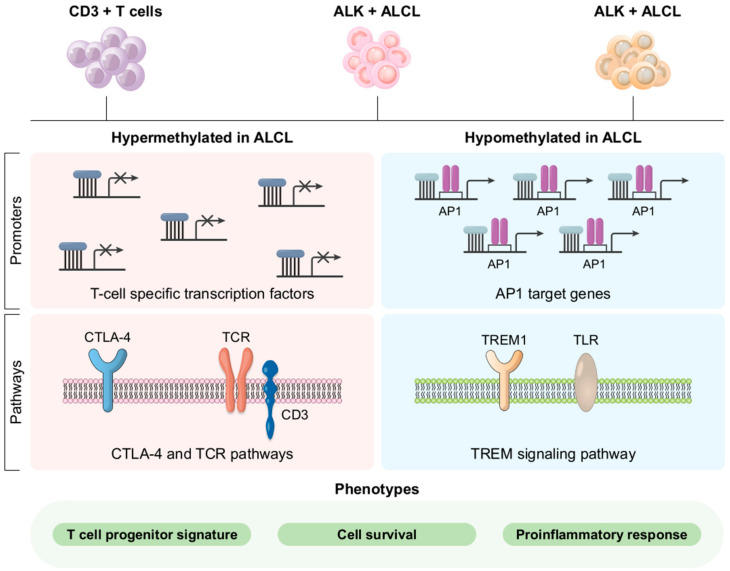
Both ALK+ and ALK−ALCL have a genome-wide DNA methylation pattern. ALCL tumor cells are similar to undifferentiated thymic progenitor cells in appearance. Major T cell activities are linked differentially to methylation genes. Epigenetic fingerprints are left by oncogenic transcription factors. Reproduced with permission from [91].

**Figure 4 cancers-14-01650-f004:**
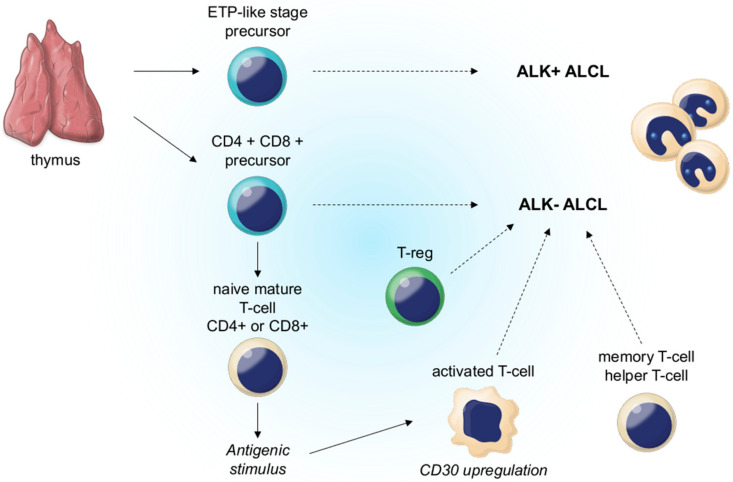
Putative cell(s) of origin of systemic ALCL. ALCL has a methylation pattern that resembles that of immature T cells/thymocytes at different stages of development. The methylation profile of ALK−ALCL is comparable to that of a double-positive (CD4+/CD8+) thymic precursor. Nonetheless, a variety of different mature T cell subtypes could be a potential precursor to this lymphoma (dashed arrows). ALK+ALCL can come from the same mature T cell subtypes as ALK−ALCL, although this is not shown for clarity. Reproduced with permission from [113].

**Figure 5 cancers-14-01650-f005:**
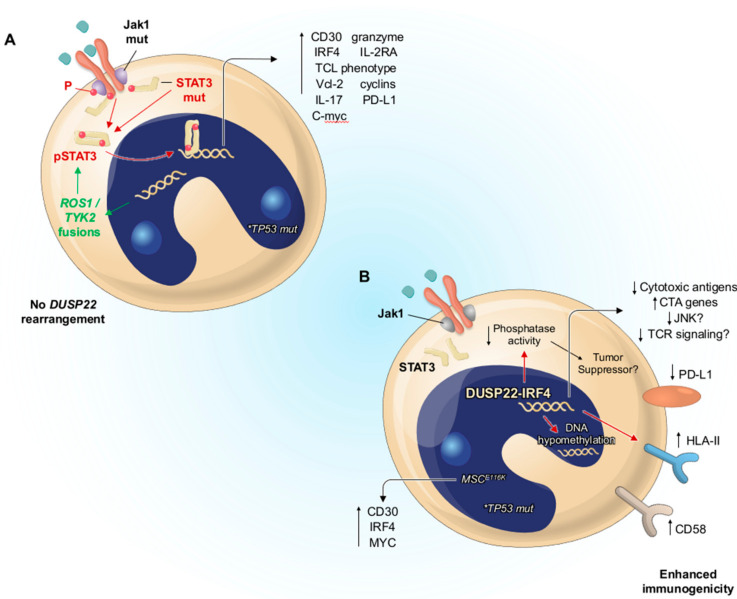
Molecular study of ALK−ALCL for inducing the pathogenic condition. (**A**) ALK-negative ALCL with mutations of JAK1 and STAT3. The activation of the JAK/STAT pathway is only found in cases without the DUSP22 rearrangement. (**B**) ALK-negative ALCL with DUSP22-IRF4 rearrangement. Reproduced with permission from [113].

**Figure 6 cancers-14-01650-f006:**
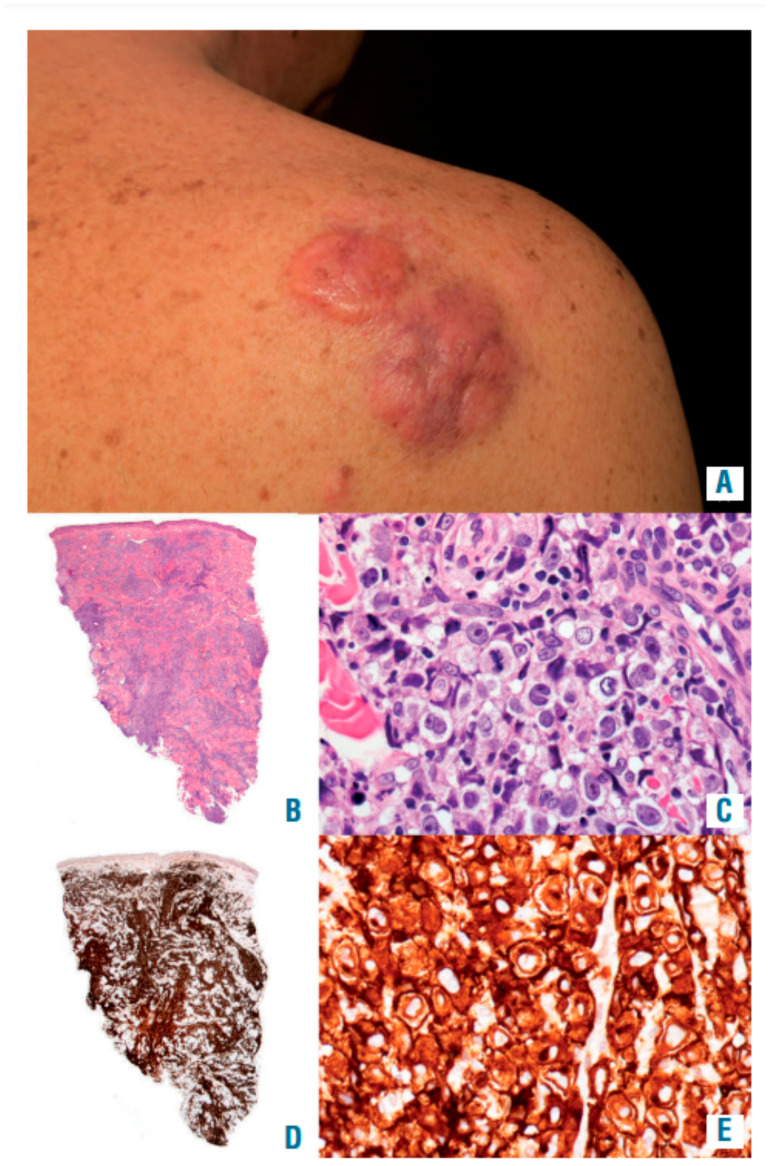
Classic, ALK-DUSp22-, TP63- primary cutaneous anaplastic large cell lymphoma. (**A**) In the scapular region, adjacent tumoral erythematous nodules resembling dermatofibrosarcoma protuberans. (**B**,**C**) Hematoxylin and eosin stain demonstrating a circumscribed infiltrate formed of organized huge lymphoid cells with missing or modest epidermotropism in the dermis. (**D**,**E**) CD30 stain showing positivity in the membrane and Golgi of the tumoral large cells. Reproduced with permission from [145].

**Figure 7 cancers-14-01650-f007:**
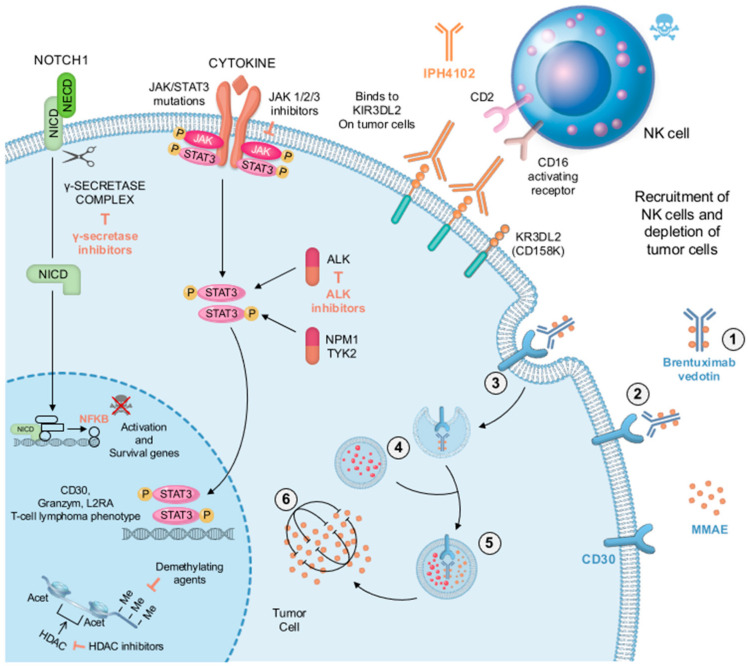
Development of new therapeutic treatment of pcALCL using brentuximab vedotin. Briefly, after combination with four monomethyl auristatin molecules (**1**), this drug binds to the surface of pcALCL (**2**). It then follows the internalizations (**3**,**4**). The resulting capsule combines with lysosomes (**5**) and then terminates in dipeptide linker proteolysis as well as free MMAE (**6**). This leads to block tubulin polymerization of the cellular cytoskeleton and prevents the development of pcALCL tumor cells. Reproduced with permission from [145].

**Figure 8 cancers-14-01650-f008:**
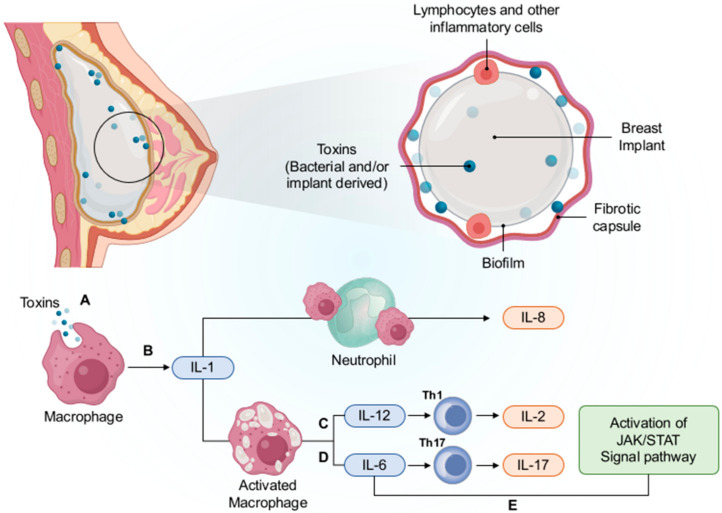
The development of immune-responsive T cells to BIA-ALCL is hypothesized. In short, macrophages will pick up bacteria (**A**). Interleukin (IL)-1 is created by the activation of cell-wall elements (**B**) and then produces IL-8 through stimulation of neutrophil and IL-12 (**C**), which promotes Th1 mutation. In addition, macrophages also cause IL-6 (**D**), which leads to Th17 cell mutation, and IL-6 also is activated through activation of the JAK/STAT signaling pathway (**E**). Reproduced with permission from [169].

**Figure 9 cancers-14-01650-f009:**
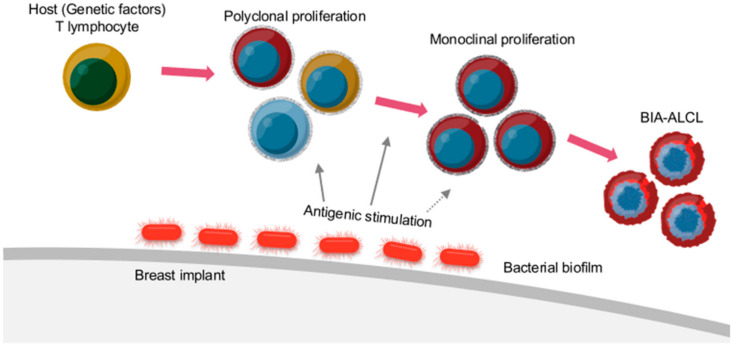
Breast implant mediated genesis of BIA-ALCL. Reproduced with permission from [183].

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
