# Peer review of "Anaplastic Large Cell Lymphoma: Molecular Pathogenesis and Treatment"

_cancers, 2022, doi:10.3390/cancers14071650_

Round 1
Reviewer 1 Report
Thank you for sending me the manuscript "Cancers 1623497" for review. This manuscript is a comprehensive review addressing "Anaplastic Large cell Lymphoma: Molecular pathogenesis and Treatment", and should be published following the review comments as follows: Line (L) 11 & L 15 suggestion that "Type or Form" rather than "Kind of", is more scientifically oriented. P 34 suggest the word "Compose about" rather than "occupy around". L 41 cells that "are prone" rather than "is prone". L 160 "and" instead of "aand", also L160 " genes myeloid" ? .L 231 "Extranodal sites" instead of" Extranodalsites". L236 "fore shadowing" instead of "foreshadowing". L321 explain "fingolimod". L332 "separate the words "rapidlygrowing". These are just examples, I would urge the authors to carefully read the manuscript and find any other errors of this type.
This is a well written review with a great deal of information related to Anaplastic Large Cell Lymphoma. I have no problem with the figures since they were all reproduced with permission from other publications.
The reference list is comprehensive. However, I would suggest that he add another (more recent) reference by Sibon D 2019 addressing " ALK-positive ALCL in adults: an individual patient data pooled analysis of 263 patients. It has relevant information that can be added to the review.
What is not addressed is a publication by Yu et al, "The clinicopathological relevance of uniform CD56 expression in anapestic large cell lymphoma: a retrospective analysis of 18 cases. "Diagnostic Pathology" 2021, https://doi.org/10.1186/s13000-020-01059-y. I believe this publication is important, as it provides relevant updated information (CD56 importance in ALCL) which will add more relevance to this manuscript, and should be included.
In summary, this is an important review regarding ALCL, and I recommend publication following the changes discussed.
Author Response
Comments and Suggestions for Authors
Thank you for sending me the manuscript "Cancers 1623497" for review. This manuscript is a comprehensive review addressing "Anaplastic Large cell Lymphoma: Molecular pathogenesis and Treatment", and should be published following the review comments as follows:
- Line (L) 11 & L 15 suggestion that "Type or Form" rather than "Kind of", is more scientifically oriented. P 34 suggest the word "Compose about" rather than "occupy around". L 41 cells that "are prone" rather than "is prone". L 160 "and" instead of "aand", also L160 " genes myeloid" ? .L 231 "Extranodal sites" instead of" Extranodalsites". L236 "fore shadowing" instead of "foreshadowing". L321 explain "fingolimod". L332 "separate the words "rapidlygrowing". These are just examples, I would urge the authors to carefully read the manuscript and find any other errors of this type.
Answer: Thank you for your suggestions. We have changed “kind” to “type” on Page 1, Line 11 and 15.
We have changed “occupy about” to “compose about” on Page 1, Line 34.
We have changed “is prone” to “are prone” on Page 1, Line 41.
We have changed “aand” to “and” on Page 5, Line 159.
We have changed “genes myeloid” to “protein myeloid” on Page 5, Line 160.
We have changed “Extranodalsites” to “Extranodal sites” on Page 7, Line 231.
Thank you for your suggesting the word “fore shadowing”, however, we decided to keep “foreshadowing” on Page 7, Line 236, because “foreshadow” is the correct spelling of this word.
The word "fingolimod" on Page 9, Line 321 is an immunomodulating medication, mostly used for treating multiple sclerosis.
We have changed “rapidlygrowing” to “rapidly growing” on Page 10, Line 332.
- This is a well written review with a great deal of information related to Anaplastic Large Cell Lymphoma. I have no problem with the figures since they were all reproduced with permission from other publications.
Answer: Thank you for your comments.
- The reference list is comprehensive. However, I would suggest that he add another (more recent) reference by Sibon D 2019 addressing " ALK-positive ALCL in adults: an individual patient data pooled analysis of 263 patients. It has relevant information that can be added to the review.
Answer: Thank you for your suggestion. According to reviewer’s suggestion, we have added the content related to the reference named “ALK-positive ALCL in adults: an individual patient data pooled analysis of 263 patients.” on Page 5, Line186-193.
“Furthermore, a pooled analysis of 263 patients demonstrated that etoposide-based induction was associated with improved 5-year PFS (83% versus 62%) and 5-year OS (93% versus 74%) using etoposide versus non-etoposide regimens. In patients ≤60 years (n=232), the respective 5-year PFS and OS were 81% versus 65%, and 92% versus 77%. However, among the 31 patients >60 years, given the additive toxicity, only six received etoposide with their induction and 25 did not. The respective 5-year PFS and OS were 100% versus 42%, and 100% versus 53% [75].”
Sibon, D.; Nguyen, D. P.; Schmitz, N.; Suzuki, R.; Feldman, A.L.; Gressin, R.; Lamant, L.; Weisenburger, D.D.; Rosenwald, A.; Nakamura, S.; Ziepert, M.; Maurer, M.J.; Bast, M.; Armitage, J.O.; Vose, J.M.; Tilly, H.; Jais, J.P.; Savage, K.J. ALK-positive anaplastic large-cell lymphoma in adults: an individual patient data pooled analysis of 263 patients. Haematologica2019, 104, e562–e565.
- What is not addressed is a publication by Yu et al, "The clinicopathological relevance of uniform CD56 expression in anapestic large cell lymphoma: a retrospective analysis of 18 cases. "Diagnostic Pathology" 2021, https://doi.org/10.1186/s13000-020-01059-y. I believe this publication is important, as it provides relevant updated information (CD56 importance in ALCL) which will add more relevance to this manuscript, and should be included.
Answer: Thank you for your suggestion. According to reviewer’s suggestion, we have the content related to the reference named “The clinicopathological relevance of uniform CD56 expression in anapestic large cell lymphoma: a retrospective analysis of 18 cases.” on Page 5, Line162-166.
“Moreover, it is worth noting that there exist some cases showing the expression of CD56. According to retrospective analysis, the proportion of systemic ALK-positive ALCL presenting CD56+ is about 20%. The cases which underwent TCR gene rearrangement detection exhibited monoclonal TCR gene rearrangement, with the same phenotype of TCR-γ+/β+/δ- [71]. ”
Yu, B.H.; Yan, Z.; Tian, X.; Shui, R.H.; Lu, H.F.; Zhou, X.Y.; Zhu, X.Z.; Li, X.Q. The clinicopathological relevance of uniform CD56 expression in anaplastic large cell lymphoma: a retrospective analysis of 18 cases. Diagn Pathol2021, 16, 1.
- In summary, this is an important review regarding ALCL, and I recommend publication following the changes discussed.
Answer: Thank you for your comments.

Reviewer 2 Report
Clear and well written review focused on the molecular pathogenesis and management of anaplastic large cell lymphoma.
Specific comments regarding the manuscript ID: cancers-1623497:
1. The paper is a review regarding the main genetic, clinical and therapeutic aspects of Anaplastic Large Cell Lymphoma.
2. This review is useful for clinicians because this lymphoma is relatively rare and includes four subgroups which have different genetic and clinical aspects which need differentiated therapeutic approaches.
3. It is a concise and well documented paper which also allows non-specialists to understand the main features of this rare lymphoma.
4. They should make an effort to focus more on the new targeted therapies based on molecular data of the four different subgroups of Anaplastic Large Cell Lymphoma.
5. The conclusions are a little bit concise. They could be more detailed facing the main arguments presented in the text.
6,7. The references are as appropriate as the tables.
Author Response
Clear and well written review focused on the molecular pathogenesis and management of anaplastic large cell lymphoma.
Specific comments regarding the manuscript ID: cancers-1623497:
- The paper is a review regarding the main genetic, clinical and therapeutic aspects of Anaplastic Large Cell Lymphoma.
Answer: Thank you for your comments.
- This review is useful for clinicians because this lymphoma is relatively rare and includes four subgroups which have different genetic and clinical aspects which need differentiated therapeutic approaches.
Answer: Thank you for your comments.
- It is a concise and well documented paper which also allows non-specialists to understand the main features of this rare lymphoma.
Answer: Thank you for your comments.
- They should make an effort to focus more on the new targeted therapies based on molecular data of the four different subgroups of Anaplastic Large Cell Lymphoma.
Answer: Thank you for your suggestion. The content about new targeted therapies has been added on Page 6, Line 202-210.
“Crizotinib, an ALK inhibitor, has shown to have very excellent effects in clinical trials, notably in the juvenile population [80, 81]. In a trial conducted by the Children's On-cology Group (COG), 21 out of 26 pediatric patients showed a complete response to crizotinib as a first-line treatment for ALK inhibition [82]. Unfortunately, patients with ALK-positive lymphoma who stopped taking crizotinib experienced a sudden relapse [83]. Several clinical trials with crizotinib, lorlatinib, and ceritinib present promising preliminary results. Despite the preliminary efficacy of ALK kinase inhibition in ALK + ALCL, resistance mutations have been discovered [84], reducing ALCL cell sensitivity to several ALK inhibitors [85].”
Gambacorti-Passerini, C.; Messa, C.; Pogliani, E.M. Crizotinib in anaplastic large-cell lymphoma. N Engl J Med2011, 364, 775–776.
Mosse, Y.P.; Lim, M.S.; Voss, S.D.; Wilner, K.; Ruffner, K.; Laliberte, J. et al. Safety and activity of crizotinib for paediatric patients with refractory solid tumours or anaplastic large-cell lymphoma: a Children’s Oncology Group phase 1 consortium study. Lancet Oncol2013, 14, 472–80.
Mosse, Y.P.; Voss, S.D.; Lim, M.S.; Rolland, D.; Minard, C.G.; Fox, E. et al. Targeting ALK With Crizotinib in Pediatric Ana-plastic Large Cell Lymphoma and Inflammatory Myofibroblastic Tumor: A Children’s Oncology Group Study. J Clin Oncol2017, 35, 3215–3221.
Gambacorti-Passerini, C.; Mussolin, L.; Brugieres, L. Abrupt Relapse of ALK-Positive Lymphoma after Discontinuation of Crizotinib. N Engl J Med2016, 374, 95–96.
Gambacorti-Passerini, C.; Farina, F.; Stasia, A.; Redaelli, S.; Ceccon, M.; Mologni, L. et al. Crizotinib in advanced, chemo-resistant anaplastic lymphoma kinase-positive lymphoma patients. J Natl Cancer Inst2014, 106, djt378.
Ceccon, M.; Mologni, L.; Bisson, W.; Scapozza, L.; Gambacorti-Passerini, C. Crizotinib-resistant NPM-ALK mutants confer differential sensitivity to unrelated Alk inhibitors. Mol Cancer Res2013, 11, 122–132.
Page 9, Line 327-330.
“Targeted therapy for ALK- ALCL is research currently. Ruxolitinib, a JAK1/2 in-hibitor, was confirm to be effective in vivo. The data suggested that cytokine receptor signaling is required for the survival of tumor cells, even in the presence of JAK1/STAT3 mutations. Thus, JAK inhibitor therapy is helpful for ALK- ALCL man-agement [128].”
Chen, J.; Zhang, Y.; Petrus, M.N.; Xiao, W.; Nicolae, A.; Raffeld, M.; Pittaluga, S.; Bamford, R.N.; Nakagawa, M.; Ouyang, S.T.; Epstein, A.L.; Kadin, M.E.; Del-Mistro, A.; Woessner, R.; Jaffe, E.S.; Waldmann, T.A. Cytokine receptor signaling is required for the survival of ALK- anaplastic large cell lymphoma, even in the presence of JAK1/STAT3 mutations. Proc Natl Acad Sci U S A2017, 114, 3975-3980.
Page 16, Line 585-596.
“Dupilumab, a fully human monoclonal antibody that targets IL-4/IL-13 signaling, has shown promise in clinical studies for severe asthma [190] and atopic dermatitis [191], and it may be worth considering for BIA-ALCL treatment. Although IL-6 has been proposed as a driving cytokine in BIA-ALCL, inhibition of the cytokine had little effect on in vitro growth. Clinical trials using tocilizumab, a humanized monoclonal anti-body that targets IL-6R, have showed promise in the treatment of chronic inflamma-tory, autoimmune, and lymphoproliferative illnesses. While IL-6 targeted therapy has not been shown to be effective in the treatment of overt cancers, it may be useful in the development of BIA-ALCL. Another strategy targets the downstream effects of IL-6 signaling, specifically JAK/STAT signaling. Ruxolitinib, a JAK1/2 inhibitor, is effective in suppressing the the proliferation of tumor cells and benefits the patients with this lymphoma [192].”
Barranco, P.; Phillips-Angles, E.; Dominguez-Ortega, J.; Quirce, S.; Dupilumab in the management of moderate-to-severe asthma: the data so far. Ther Clin Risk Manag2017, 13, 1139-1149.
Xu, X.; Zheng, Y.; Zhang, X.; He, Y.; Li, C.; Efficacy and safety of dupilumab for the treatment of moderate-to-severe atopic dermatitis in adults. Oncotarget2017, 8, 108480-108491.
Crescenzo, R.; Abate, F.; Lasorsa, E. et al. Convergent mutations and kinase fusions lead to oncogenic STAT3 activation in anaplastic large cell lymphoma. Cancer Cell2015, 27, 516-532.
- The conclusions are a little bit concise. They could be more detailed facing the main arguments presented in the text.
Answer: Thank you for your suggestion. We have summarized the conclusion on Page 16, Line 598-612.
“Anaplastic large cell lymphoma is an uncommon form of T cell lymphomas. It is crucial to know all the subtypes of ALCL in consequence of distinct mechanisms, treatment, and prognosis. In addition, aberrant histologic variants or cases with a "null" phenotype can be overlooked or confused with a variety of other hematopoietic and non-hematopoietic malignancies. Therefore, it is necessary to be aware of the clinical features of ALCL. The recognition of alterations involving genes and JAK/STAT signaling pathway can assist us to stratify these cases into those with a better or worse prognosis and optimize the treatment. Currently, the anthracycline-based chemotherapy regimen is the most widely used therapy throughout the world. The addition of Etoposide to CHOP regimen has been confirmed that it can improve the PFS and OS significantly. Besides, the application of targeted therapies based on molecular mechanism have been studied, and Brentuximab vedotin as well as other inhibitors are helpful for treatment. The understanding of the molecular pathogenesis and management of anaplastic large cell lymphoma will bring on a profound and positive impact on the study of this disease, which may help to make a progress in the field of oncology.”
6,7. The references are as appropriate as the tables.
Answer: Thank you for your comments.
This manuscript is a resubmission of an earlier submission. The following is a list of the peer review reports and author responses from that submission.